# The Association between Parental Child Vaccination Refusal Rate and the Impact of Mass Vaccination against COVID-19 in Kazakhstan: An Interrupted Time Series Analysis with Predictive Modelling of Nationwide Data Sources from 2013 to 2022

**DOI:** 10.3390/vaccines12040429

**Published:** 2024-04-17

**Authors:** Madina Abenova, Askhat Shaltynov, Ulzhan Jamedinova, Erlan Ospanov, Yuliya Semenova

**Affiliations:** 1Department of Biostatistics and Epidemiology, Semey Medical University, Semey 071400, Kazakhstan; madina.abenova@smu.edu.kz (M.A.); askhat.shaltynov@smu.edu.kz (A.S.); ulzhan.jamedinova@smu.edu.kz (U.J.); erlan.ospanov@smu.edu.kz (E.O.); 2School of Medicine, Nazarbayev University, Astana 010000, Kazakhstan

**Keywords:** vaccination refusal, routine vaccine hesitancy, vaccine-preventable diseases, mandatory vaccination, vaccination confidence, COVID-19

## Abstract

Despite well-established evidence supporting vaccination efficacy in reducing morbidity and mortality among infants and children, there is a global challenge with an increasing number of childhood vaccination refusals. This issue has intensified, especially during the COVID-19 pandemic. Our study aims to forecast mandatory childhood vaccination refusal trends in Kazakhstan until 2030, assessing the impact of mass COVID-19 vaccination on these rates. Utilizing annual official statistical data from 2013 to 2022 provided by the Ministry of Health of Kazakhstan, the study reveals a significant surge in refusals during the pandemic and post-pandemic periods, reaching record levels of 42,282 cases in 2021 and 44,180 cases in 2022. Notably, refusal rates sharply rose in specific regions, like Aktobe (13.9 times increase) and Atyrau (4.29 times increase), emphasizing the need for increased public healthcare attention in these areas. However, despite a decade of data, our forecasting analysis indicates a lack of volatility in childhood vaccine refusal trends for all vaccine types up to 2030, highlighting the statistical significance of the obtained results. The increasing trend in vaccine refusals underscores the necessity to enhance crisis response and support health initiatives, particularly in regions where a substantial rise in refusals has been observed in recent years.

## 1. Introduction

Vaccination is widely recognized as a highly efficacious public health measure that significantly mitigates the morbidity and mortality associated with vaccine-preventable diseases. Through comprehensive immunization efforts, an estimated 120 million deaths from various infectious diseases in children born between 2000 and 2030 have been averted, with a 95% confidence interval ranging from 93 to 150 million [1]. Moreover, vaccination has been instrumental in eradicating certain diseases, such as poliomyelitis, from the population altogether. It is also important to note the nonspecific impact of mandatory childhood vaccination on reducing all-cause mortality. Several studies have suggested broader beneficial effects of these vaccines through the induction of innate immune memory, which has a beneficial effect on survival, especially in children with weakened immune systems. Therefore, decreased vaccination coverage and increased hesitancy could lead to serious public health problems in the long term [2,3,4,5,6].

Immunization plays a critical role in advancing progress towards 14 out of the 17 Sustainable Development Goals, as outlined by the Global Vaccine Action Plan’s target of achieving a minimum of 90% vaccination coverage by 2020 [7]. While several countries have made noteworthy strides in meeting these targets through the implementation of national and regional immunization programs, the global COVID-19 pandemic has had a significant impact on immunization rates, with a decline observed from 2019 to 2021 in comparison to the period from 2015 to 2019 [8]. Contributing to this decline are increasing misinformation and hesitancy about vaccines, which have emerged as major barriers to childhood immunization and are recognized as one of the top 10 global public health threats [9,10]. To effectively address this challenge and prevent outbreaks, public health officials need to employ targeted strategies aimed at restoring and enhancing immunization programs [11]. In this regard, the newly proposed 2030 Immunization Agenda (IA2030) presents a global vision and strategy that seeks to sustain and expand vaccination coverage, while improving access to vaccines to combat vaccine-preventable diseases [12,13].

The Republic of Kazakhstan, a Central Asian country with a population exceeding 19 million, has experienced a notable positive trend in the birth rate of children from 2013 to 2022, with an increase from 387,256 to 403,893. Despite annual vaccination efforts that reach approximately 5 million individuals, including 1.3 million children, statistical data reveals a decline in the coverage of preventive vaccinations among children over the past decade. Specifically, coverage rates for the diphtheria, tetanus, and pertussis (DTaP) vaccine declined from 95.6% to 88.3%, the oral polio vaccine (OPV) from 97.9% to 88.3%, and the measles, mumps, and rubella (MMR) vaccine from 99.5% to 92.9% [14]. Contributing factors to this trend include the global COVID-19 pandemic, limited knowledge about vaccines and vaccination sources, and the presence of distrust in the healthcare system due to the lack of communication skills among medical personnel [15]. In the beginning of February 2021, a phased mass vaccination of the population against COVID-19 commenced. The process commenced with individuals falling into high-risk categories, including medical professionals working in infectious hospitals, emergency medical services, intensive care units, primary medical care providers, personnel of reception rooms, sanitary and epidemiological service staff, educators, students, and individuals with chronic illnesses. Subsequently, the vaccination campaign expanded to encompass the broader population [16]. Against this backdrop, the prevalence of parents expressing hesitation or outright refusal towards vaccination is steadily growing on an annual basis. The fact that Kazakhstan is among the top three countries with the highest measles incidence (13,677 cases) and an increase in pertussis cases (393 cases in 2023, which is 130 times higher compared to the previous year) underscores the urgent need for taking immediate measures to ensure access to information for the population and the development of preventive measures [17,18].

The immunoprophylaxis of infectious diseases in Kazakhstan follows the National Immunization Schedule (NIS) and the vaccination schedule for epidemic indications. This includes administration of all mandatory vaccinations, such as the bacille Calmette–Guérin (BCG) vaccine, the hepatitis B vaccine (HBV), the diphtheria, tetanus, pertussis, inactivated poliovirus, haemophilus influenzae type b, and hepatitis B combination vaccine (DTaP-HBV-IPV/Hib vaccine), the diphtheria, tetanus, pertussis, inactivated poliovirus, and haemophilus influenzae type b combination vaccine (DTaP-IPV/Hib vaccine), the OPV, the pneumococcal conjugate vaccine (PCV), and the MMR (measles, mumps, and rubella) vaccine, targeting 11 infections including tuberculosis, diphtheria, poliomyelitis, measles, mumps, rubella, tetanus, whooping cough, hepatitis B, hemophilic infection, and pneumococcal infection. Since the beginning of 2016, combined forms of vaccines, such as the DTaP-HBV-IPV/Hib vaccine and the DTaP-IPV/Hib vaccine, have been introduced, replacing the use of monovalent vaccines that were utilized before this period. These vaccinations are provided free of charge and are funded by the national budget, commencing from 1 to 4 days of life and continuing up to 6 years of age [19,20,21]. According to the vaccination procedure in Kazakhstan, vaccinations are administered subsequent to obtaining informed consent from parents or legal representatives, either in paper or electronic form. The procedure is not carried out without obtaining informed consent [21].

The existing scientific literature highlights a notable dearth of data pertaining to trend forecasting of vaccine-preventable infections in Kazakhstan, thereby hindering the ability of public health authorities to formulate optimal vaccine policies, programs, and investments. Therefore, the aim of this study is to evaluate the past trends in refusals of mandatory childhood vaccination in Kazakhstan for the period 2013–2022, make forecasts until 2030, and evaluate the impact of mass vaccination against COVID-19 on the refusal rates of childhood vaccinations.

## 2. Materials and Methods

### 2.1. Study Design and Data Sources

This research constitutes a retrospective study. Official statistical data, supplied by the Committee of Sanitary and Epidemiological Control of the Ministry of Health of the Republic of Kazakhstan (CSEC of the MoH RK) and the Scientific and Practical Center for Sanitary and Epidemiological Expertise and Monitoring Branch of the Republican State Enterprise on the right of economic use “National Center for Public Health” (RSE NCoPH) of the MoH RK, were employed throughout the investigation. These data specifically address parental/guardian refusals for mandatory vaccinations among children aged 0–14 years, along with the incidence rates of infectious and parasitic diseases resulting from both monovalent and combination vaccines included in the NIS of the MoH RK. The overall number of children aged 0–14 years in the RK, segmented by regions and cities of republican significance, was derived from data provided by the Bureau of National Statistics of the Agency for Strategic Planning and Reforms of the Republic of Kazakhstan (BNSASPR of the RK).

For the purpose of facilitating international comparisons in the fields of child health, education, and social well-being, which is a common practice in many countries, a uniform age range of children up to 15 years old was chosen.

Over the past decade, spanning from 2013 to 2022, data has been scrutinized concerning parental/guardian refusals of vaccination in the regions of the RK. The data was categorized into three temporal intervals: 2013–2016, 2017–2019, and 2020–2022, aiming to investigate the dynamics of vaccination refusals.

Data collected for analysis was gathered in accordance with Form No. 4 of the reporting documentation in the healthcare sector. This form was approved by the Order of the MoH RK on 22 December 2020 (No. KR DSM-313/2020). The form contains information on the total number of children who have not received vaccinations, contraindications (including both temporary and permanent), and the number of children not vaccinated due to parental refusal across regions and the RK as a whole [22]. Medical professionals, including nurses and statisticians, fill out this form based on the interaction between healthcare organizations, depending on the level of medical care provided (village, district, city, region, and central). This includes all outpatient and polyclinic organizations in the healthcare system, non-governmental medical institutions, government authorities, and other organizations providing medical care to children and adults. This encompasses rural feldsher-obstetric points, district and city hospitals or clinics, regional health departments, and the Scientific and Practical Center for Sanitary and Epidemiological Expertise and Monitoring branch of the RSE NCoPH. A monthly report is submitted to the CSEC of the MoH RK based on primary accounting data, such as the Vaccination Card (Form No. 065/u), Vaccination Record-Journal (Form No. 066/u), Exchange Card for Pregnant Women and Maternity (Form No. 048/u), Outpatient Patient Medical Card (Form No. 052/u), and other primary accounting documents approved by the acting order of the MoH RK [23].

However, despite the mandatory nature of childhood vaccination according to the vaccination schedule, some parents may not refuse vaccination but may skip or postpone the process, failing to adhere to established deadlines. Likely, such cases will not be reflected in statistical data, and the number of parents expressing doubts or refusing childhood vaccination may increase.

The calculation of standardized vaccination refusal rates across regions was conducted using the following formula:Standardized prevalence rate of vaccination refusal = Sum of vaccination refusals for all vaccine types/Annual average number of children aged 0–14 (calculated separately for each year)) × 100,000

The overall indicators of vaccination refusals were generated by aggregating individual refusal rates, considering various vaccine types (the BCG vaccine, the OPV, the MMR vaccine, the DTP vaccine, the pneumococcal vaccine, the DTaP-HBV-IPV/Hib vaccine, and the DTaP-IPV/Hib vaccine). In other words, the data in the article revealed the overall summarized indicator of refusals from childhood vaccinations, depending on refusals for each type of vaccine established by the National Immunization Schedule separately. Due to the absence of comprehensive statistical data on the annual average number of children aged 0–14 and their gender distribution, the total number of children was calculated individually for each year and region. As the data were divided into three temporal intervals, average values of individual standardized prevalence rates of vaccination refusal were utilized for each year based on the region in Kazakhstan. In 2022, significant changes occurred in the territorial division, with the emergence of three new regions: Abay, Ulytau, and Zhetisu, formed by separating them from the East Kazakhstan, Karaganda, and Almaty regions. However, statistical data for 2022, reflecting these changes, were not collected. Therefore, all data were considered without differentiation by regions in the year 2022.

### 2.2. Statistical Analysis

Three software programs were utilized for data analysis: WinPepi version 11.65 (developed by Professor Y.H. Abramson at Hebrew University and Hadassah Faculty of Medicine), Statistical Package for Social Sciences (IBM SPSS Statistics, New York, NY, USA) version 20.0, and ArcMap version 10.8.1 by Environmental Systems Research Institute (ESRI, Redlands, CA, USA). WinPepi was employed to compute the average change per annum along with a confidence interval (CI). IBM SPSS Statistics was used to perform forecasting analysis and analyze time series data. The Expert Modeler function within IBM SPSS Statistics played a pivotal role in identifying the optimal epidemiological model for predictive analysis in the study. To forecast the number of vaccine refusals until 2030, data from the period 2013 to 2022 were utilized, generating forecasts with a 95% CI. Parameters of the optimal model were extracted. The analysis of time series with monthly periodicity involved creating graphs of vaccine refusal numbers. The intervention period was selected as February 2021, corresponding to the onset of mass vaccination against COVID-19. Percentage point changes (PPCs) were calculated to assess differences between actual and counterfactual refusal rates in an interrupted time series analysis. All conducted tests were considered significant at a significance level of *p* = 0.05.

To visualize regional changes in the dynamics of parental/guardian vaccination refusals per 100,000 children aged 0–14, maps of Kazakhstan were created for three temporal periods: 2013–2016, 2017–2019, and 2020–2022 using ArcMap version 10.8.1 by ESRI.

### 2.3. Ethical Considerations

The study was approved by the Ethical committee of the Semey Medical University (Semey, Kazakhstan (protocol #8 dated 24 May 2022).

## 3. Results

In total, data from more than 50 million children under the age of 15 were exported and analyzed from 2013 to 2022. The overall trend of parental/guardian/caregiver vaccine refusals indicates a 40.19% increase (95% CI 14.64–71.44%). In most cases, there is a general trend of growth in the first half of the period (2013–2016), followed by stabilization or a reduction in refusal cases. Concerning certain vaccines (e.g., BCG and hepatitis B), a significant increase in vaccine refusals was observed, while for others (the DTaP-HBV-IPV/Hib and DTaP-IPV/Hib vaccines), there is a decline. The highest refusal rates are noted for the following vaccines: BCG with an average change per annum of 48.8% (95% CI 20.56–83.64%), hepatitis B with 42.75% (95% CI 17.67–73.17%), and DTaP with 42.18% (95% CI 14.46–76.63%). However, despite the slight increase in refusals for vaccines such as DTaP and Hib, administered separately until 2016, there is a significant decrease in parental mistrust toward the DTaP-HBV-IPV/Hib and DTaP-IPV/Hib combination vaccines, with a relative reduction of 3.38% (95% CI −16.08–11.26%) and 3.99% (95% CI −16.25–10.07%), respectively. Despite the inclusion of the pneumococcal vaccine since 2016, the parental attitude toward this vaccine among children under 15 shows a decrease in refusal cases, with a relative reduction of 1.09% (95% CI −12.30–11.54%). Two temporal intervals with the highest growth rates of vaccine refusals before the pandemic period (2016–2017) were observed, with over 40 thousand refusal cases and during the pandemic period starting from 2020, showing a sharp increase by 25.7% compared to 2019, and a subsequent growth to 44 thousand refusal cases in 2022 (Table 1).

The data on the prevalence of vaccine refusal among parents of children aged 0–14 in various regions of Kazakhstan during the periods 2013–2016, 2017–2019, and 2020–2022 were analyzed (Figure 1).

The overall trend in Kazakhstan indicates that the general level of vaccine refusal has increased by 2.62 times over the past 10 years. Despite some fluctuations in different regions, the rise in refusals in 2016 predominates in most areas (Figure 1). However, from 2017 to 2019, the overall trend across Kazakhstan suggests a decrease in vaccine refusals compared to the years 2013–2016 (Figure 1A,B). A significant increase in maximum refusal levels in 2020–2022 was observed across all 17 regions and cities of republican significance (Astana, Almaty, Shymkent) (Figure 1C). A consistently high level of vaccine refusals is maintained in the Turkistan and Mangystau regions. Nevertheless, the overall refusal rates in the Turkistan region showed negative trends in 2020–2022 compared to 2017–2019.

It is worth noting the sharp increase in the refusal level in the Aktobe region by 13.9 times since 2013 (from 123.5 to 1726.1) and in the Atyrau region by 4.29 times (from 244.1 to 1048.6). The lowest refusal rates in all three periods were found in the East Kazakhstan, Zhambyl, and Kyzylorda regions. However, in the Kyzylorda region, despite comparatively low refusal rates compared to other regions, a significant increase in rates has been observed over the last 10 years (from 45.3 to 200.3). Stable rates were maintained in the North Kazakhstan region from 2017 to 2022. A decrease in rates was also observed in three regions from 2017 to 2022: Almaty (from 246.6 to 220.4), West Kazakhstan (from 579.2 to 481.1), and Zhambyl region (from 155.4 to 141.2).

An analysis was conducted to forecast the refusal of mandatory childhood vaccination until 2030, considering various types of vaccines included in the NIS of the RK: the BCG vaccine, OPV, the hepatitis B vaccine, the MMR vaccine, the PCV, the DTaP-HBV-IPV/Hib vaccine, and the DTaP-IPV/Hib vaccine (refer to Table 2 and Figure 2). According to the presented forecast data, vaccination refusals among children under the age of 15 vary depending on the type of vaccines, ranging from 68.7 (95% CI: −70.4–207.8) to 181.5 (95% CI: −85.7–448.7) in 2026. Similar indicators for 2030 are also provided, but with different confidence intervals, ranging from 68.7 (95% CI: −128.0–265.4) to 181.5 (95% CI: −196.2–559.3). Despite the consistent statistics, a statistically significant trend is observed for all vaccine types.

Figure 2 serves as a visual representation of the data from Table 2, illustrating the dynamics of refusals of mandatory childhood vaccination in Kazakhstan, considering various vaccine types included in the NIS of the RK. The graph indicates that the number of refusals fluctuates from 2013 to 2022. Particularly significant increases in refusals occurred in 2016 for the hepatitis B and MMR vaccines, in 2017 for the BCG vaccine and the OPV, and in 2019 for other vaccine types. Forecasts suggest a continuation of the trend of vaccine refusals in the future. In the absence of appropriate precautions, recurrent outbreaks of diseases are possible, leading to a significant increase in the disease burden.

A series of interrupted time series analyses were conducted to investigate changes in the number of refusals associated with various mandatory childhood vaccines before and after the phased mass vaccination against COVID-19 was implemented in Kazakhstan starting from February 2021 (see Table 3). The evaluation of variability, predicted by the model (stationary R-squared), identified the most suitable models, the number of predictors, and the extent of the intervention’s impact. The analysis results indicate a significant increase in refusals for vaccines such as the BCG vaccine, the OPV, the hepatitis B vaccine, the MMR vaccine, the PCV, and the DTaP-IPV/Hib vaccine, while refusals for the DTaP-HBV-IPV/Hib vaccine were insignificant. The observed stationary R^2^ confirms the substantial contribution of this intervention to the overall outcome.

Figure 3A,B vividly depict the temporal trends of vaccine refusals in Kazakhstan based on monthly data from 2013 to 2022. These findings complement the results presented in Table 3, which highlights a significant increase in the number of refusals for mandatory vaccination during the COVID-19 pandemic and the onset of mass vaccination of the population. This dataset illustrates the yearly instances of refusals for compulsory childhood vaccination in Kazakhstan. The numbers reveal a distinct rise in vaccine refusals over the observed period, notably peaking in 2016 at 43,949 cases. Although there was a marginal decline in 2017, the figures sustained an elevated level, displaying fluctuations in the subsequent years. Noteworthy is the increase observed in 2020, with 35,718 refusals, and this pattern persisted in 2021 (42,282 cases) and 2022 (44,180 cases).

## 4. Discussion

Over the past decade, there has been a significant increase in the number of parents inclined to refuse or delay the scheduled immunization of their children, posing a threat to both children’s health and society as a whole. Studying the current situation and predicting trends in parental vaccine refusal regarding vaccine-preventable infectious diseases will enable the development of targeted programs to enhance awareness and expand vaccination coverage among children. A previous study in Kazakhstan identified a lack of published research aimed at identifying factors influencing parental decisions regarding mandatory childhood vaccination, including prognostic aspects of this topic [24].

Our study aims to forecast the trends in refusals of mandatory childhood vaccination in Kazakhstan until 2030 and evaluate the impact of mass vaccination against COVID-19 on the refusal rates of childhood vaccinations. The conducted study revealed a substantial increase in vaccine refusals during the observed period. The peak of this rise occurred in 2016, with the number of refusal cases reaching 43,949. Despite a slight decrease and variability in the subsequent years from 2017 to 2019, exceptionally high values were recorded during and after the pandemic period: 42,282 cases in 2021 and 44,180 cases in 2022. There has also been a sharp rise in refusal rates in specific regions of Kazakhstan, notably in the Aktobe region (by 13.9 times) and Atyrau region (by 4.29 times). In accordance with the findings of previous research, regional disparities are observed in the western and southern regions of the country, where vaccine refusal continues to increase for several reasons, including low levels of awareness, religious beliefs, and distrust of vaccines. In the Mangystau region, the percentage of informed parents is on average 10–15% lower than in other regions. Additionally, it has been noted that fundamental distrust of Kazakhstani healthcare is most prevalent among respondents from the Turkestan (27.4%) and Aktobe (25.7%) regions, underscoring the need for increased attention from healthcare authorities in these areas [25]. The results of the predictive analysis reveals an absence of volatility in the trends of children’s vaccine refusals in the upcoming years (until 2030), as evidenced by the statistical significance of the obtained data. These findings contradict the initial hypothesis of the study, which suggested an anticipated increase in vaccine refusal rates in the future. This may be linked to the notion that the potential for growth in vaccine refusals has already peaked. However, despite this, it is imperative to implement measures to reduce this level, as without active interventions and the implementation of appropriate measures, the level of vaccine refusals will persist without diminishing.

Notwithstanding scientifically substantiated evidence of the effectiveness of vaccination in reducing morbidity and mortality among infants and children, a serious global problem persists with the increasing cases of refusal of childhood vaccination, which has escalated even further, especially during the COVID-19 pandemic [26]. This issue is particularly acute in the former Soviet Union countries, despite the provision of free vaccinations in these regions [27]. The high level of refusals leads to periodic outbreaks of diseases, especially in 2015 and 2019, such as measles and pertussis. Despite preventive measures taken in previous years, outbreaks of infectious diseases in Kazakhstan recurred in 2023, reaching peak levels.

The previous year in Kazakhstan witnessed a surge in measles and pertussis cases, reaching record levels for the World Health Organization’s European region (13,677 measles cases and 393 pertussis cases). Notably, 70% of measles cases and 90% of pertussis cases were recorded among children who had not received vaccinations [28,29]. It is worth emphasizing that preventive measures implemented in 2020, such as lockdowns and mandatory mask-wearing, significantly limited access to medical services, including routine vaccinations. This resulted in a decline in vaccination coverage and an increase in the number of unvaccinated and under-vaccinated children. The findings from the study led by Causey and her colleagues identified disruptions in the coverage of routine immunizations for children’s third doses of the diphtheria, tetanus, and pertussis vaccine (DTaP3) and the first dose of the measles-containing vaccine (MCV1) in at least 85 countries in 2020. These disruptions affected approximately 80 million children under the age of 1 worldwide due to the COVID-19 pandemic [30,31]. Similar results were reported by researchers, including Evans et al., highlighting a decline in mandatory vaccination rates in the first year following the pandemic [32].

There are several factors influencing the level of vaccination coverage, including a shortage of human resources, limited knowledge among the population and healthcare workers, distrust in medical professionals, and a lack of interpersonal communication skills in the medical environment. The growing interest in information through digital platforms has led many to seek advice on social networks, especially during the pandemic, which, in turn, became a source of negative messages and misinformation [33]. The active advocacy efforts of the anti-vaccination movement have a particularly negative impact, especially among parents who have doubts due to insufficient knowledge about vaccines and their significance. Under the influence of unfounded arguments from anti-vaccine proponents, these parents experience uncertainty and fear, affecting their trust in healthcare workers and serving as the basis for the decision to refuse vaccination [34]. This phenomenon further complicates the issue of insufficient awareness and trust in vaccination, creating new challenges for the public healthcare system.

The results of a cross-sectional study in Kazakhstan indicate negative attitudes toward vaccination among parents whose children have already been vaccinated. Out of 239 participants (7.8%) representing this group, some had previously refused vaccination based on their own convictions. Thus, parental consent to vaccinations does not always imply a positive attitude toward vaccines [35].

Various countries are employing successful approaches and measures to address vaccine hesitancy and uncertainty. In a systematic review conducted by Jain et al., the statistical significance and effectiveness of interventions targeting caregivers in low- and middle-income countries were highlighted [36]. Researchers, including Atkinson et al., emphasize the efficacy of digital technologies in enhancing vaccine utilization and completing vaccination series compared to non-digital interventions [37]. However, long-term strides and efforts in the field of vaccination have been disrupted due to the COVID-19 pandemic, resulting in a decline in vaccination coverage to levels reminiscent of 15 years ago [32]. Findings from the analysis of data from 170 countries and territories by Shet et al. indicate a partial recovery in the provision of mandatory vaccination services worldwide post-COVID-19 pandemic [38]. Substantial efforts and measures are required to restore previous achievements. Monitoring and assessing recovery, as well as developing strategies for catch-up vaccination against most infectious childhood diseases, including measles and pertussis, are crucial [31]. Collaborative efforts of public health authorities worldwide based on IA2030 are aimed at addressing these challenges over the next decade [13].

Additionally, it is important to highlight the significance of a key initiative—countering misinformation regarding vaccination. This involves reassessing methods for monitoring and responding to misinformation, along with expanding strategies for disseminating accurate information [33]. The emphasis should be on the safety of vaccine formulations, findings from scientific research and clinical trials, various vaccine types, and the countries of manufacture. Improving educational outreach among healthcare professionals, young parents, and prospective parents is crucial, achieved through the creation of videos, written publications, expert interviews, and podcasts distributed across diverse social platforms (Instagram, YouTube, Telegram, TikTok, Facebook), as well as utilizing traditional media outlets (television, print). Forecasting the situation and systematically monitoring changes in public opinion about childhood vaccination are also essential components of addressing this issue.

This study has several limitations that need to be considered. Firstly, it faces a limitation in accessing significant forecasting data. Specifically, the lack of information on vaccine refusal cases for the year 2023, due to the complexity of nationwide statistics collection and the absence of standardized data, may impact the study at the time of article writing. Secondly, the presence of missing data associated with some parents, due to their uncertain vaccination status (neither refusing nor adhering to the vaccination schedule), may distort the complete picture of vaccine refusals. Nevertheless, we utilized all available official data sources. Thirdly, this study is not aimed at establishing a cause-and-effect relationship between vaccine refusal and coverage—this task was not within the scope of our research objectives.

However, despite these limitations, according to our data, this is the first study aimed at forecasting refusals of mandatory childhood vaccination in Kazakhstan and analyzing the impact of mass COVID-19 population vaccination on parents’ refusal rates for vaccinating children under the age of 15. The study has several strengths, including broad coverage of children up to 15 years for all major vaccines included in the National Immunization Schedule of Kazakhstan, as well as the utilization of data from the past 10 years.

## 5. Conclusions

The overall trend of vaccine refusal by parents indicates an increase by 40.19% (95% CI 14.64–71.44%). The highest refusal rates are observed for several vaccines, with the BCG vaccine showing a relative increase of 48.8% (95% CI 20.56–83.64%) and the hepatitis B vaccine experiencing a growth of 42.75% (95% CI 17.67–73.17%). Two distinct periods of high refusal rates are identified: one before the onset of the pandemic (2016–2017), with over 40 thousand cases of refusal, and another during the pandemic in 2020, showing a sharp increase of 25.7% compared to 2019, and a subsequent rise to 44 thousand cases of refusal in 2022. A significant increase in refusal rates was observed in all 17 regions and cities of republican significance (Nur-Sultan, Almaty, Shymkent) from 2020 to 2022, indicating a link between the refusal of mandatory childhood vaccination and the COVID-19 pandemic. A consistently high level of refusals is maintained in the Turkistan and Mangystau regions. Notably, there is a sharp increase in refusal rates in the Aktobe region by 13.9 times since 2013 (from 123.5 to 1726.1) and in the Atyrau region by 4.29 times (from 244.1 to 1048.6). The lowest refusal rates are found in the East Kazakhstan, Zhambyl, and Kyzylorda regions. However, despite having data for the past 10 years, our conducted forecasting analysis suggests an absence of volatility in the trend of childhood vaccine refusals for all vaccine types in the coming years (up to 2030), demonstrating the statistical significance of the obtained results.

Therefore, the overall increasing trend of vaccine refusals among children in Kazakhstan emphasizes the need to enhance efforts for effective crisis response and support fundamental health initiatives during and after crises, particularly in regions of Kazakhstan where a significant rise in refusals has been observed in recent years. It is also crucial to improve preventive measures related to catch-up vaccination, especially for the majority of infectious childhood diseases. This comprehensive set of measures is an integral component for increasing vaccination coverage, preventing the spread of infections, and fostering collective immunity in society.

In future research, special attention should be given to forecasting analyses and systematic monitoring of changes in public opinion regarding childhood vaccination. Additionally, studies should be conducted to identify the cause-and-effect relationship between vaccine refusal factors and vaccination coverage, considering regional peculiarities, and actively participating in public awareness campaigns.

## Figures and Tables

**Figure 1 vaccines-12-00429-f001:**
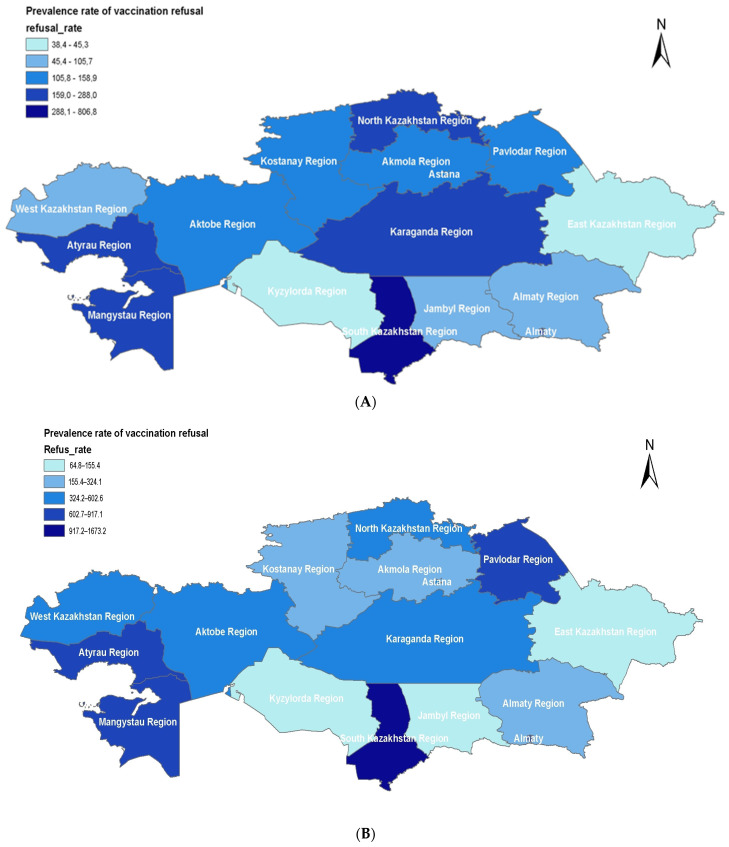
Prevalence rate of vaccination refusal among parents/caregiver of children aged 0–14 years per 100,000 population, across regions of Kazakhstan in 2013–2016 (**A**), 2017–2019 (**B**), and 2020–2022 (**C**).

**Figure 2 vaccines-12-00429-f002:**
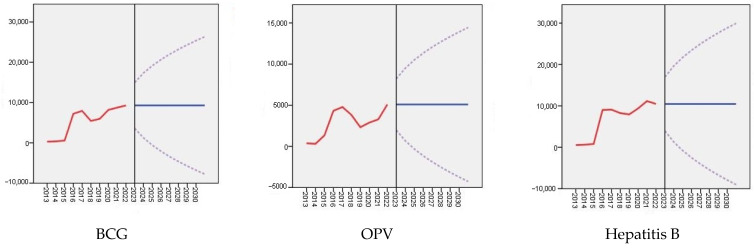
The observed and projected refusals for mandatory childhood vaccination in Kazakhstan, with respect to vaccine types.

**Figure 3 vaccines-12-00429-f003:**
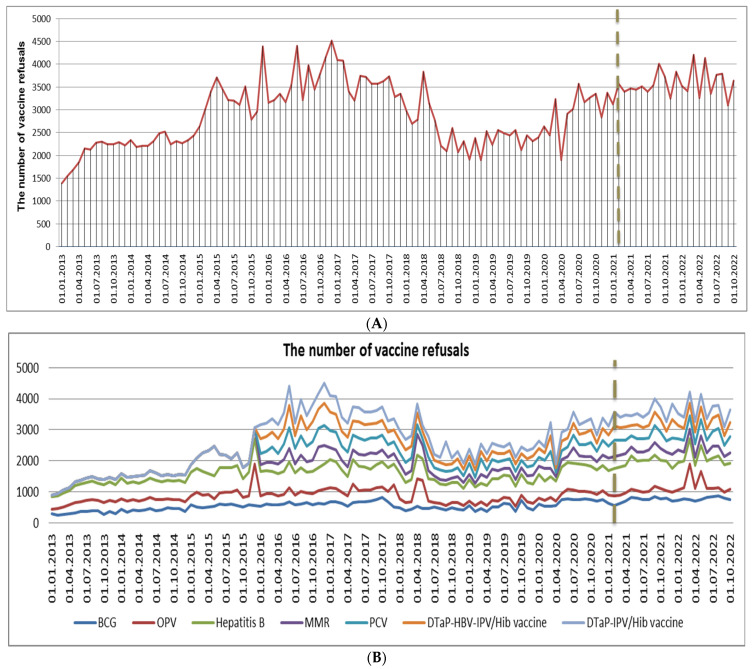
The total number of routine vaccine refusal in Kazakhstan (**A**) and with respect to vaccine type (**B**) from 1 January 2013 to 1 October 2020. The gray dashed lines indicate the commencement of the phased mass vaccination of the population against COVID-19 in Kazakhstan, starting from the beginning of February 2021.

**Table 1 vaccines-12-00429-t001:** The rate of parental/guardian/caregiver vaccine refusals for mandatory vaccination among children aged 0–14 from 2013 to 2022.

Types of Vaccines	2013	2014	2015	2016	2017	2018	2019	2020	2021	2022	Average Change per Annum	CI	Dynamic Time Series
BCG	304	358	563	7224	7929	5447	5951	8178	8749	9289	48.8%	20.56–83.64%	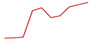
OPV	377	308	1334	4323	4787	3828	2327	2892	3283	5093	28.82%	7.16–54.85%	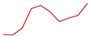
DTaP	413	454	660	-	-	-	-	-	-	-	-	-	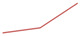
Hepatitis B	541	622	800	9019	9101	8247	7951	9404	11,169	10,470	42.75%	17.67–73.17%	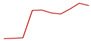
HIB	421	452	660	-	-	-	-	-	-	-	-	-	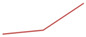
MMR	162	246	375	4505	4357	3144	2790	3483	4231	3965	42.18%	14.46–76.63%	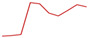
PCV	-	-	-	6238	5794	3743	3555	4270	5413	5637	−1.09%	−12.30–11.54%	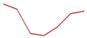
DTaP-HBV-IPV/Hib vaccine	-	-	-	6611	5809	3532	3096	3949	4970	5125	−3.38%	−16.08–11.26%	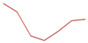
DTaP-IPV/Hib vaccine	-	-	-	6029	5306	3488	2745	3542	4467	4601	−3.99%	−16.25–10.07%	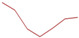
Total	2218	2440	4392	43,949	43,083	31,429	28,415	35,718	42,282	44,180	40.19%	14.64–71.44%	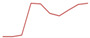

Bacille Calmette–Guérin (BCG); oral polio vaccine (OPV); diphtheria, tetanus, and pertussis (DTaP); hepatitis B vaccine (HBV); haemophilus influenzae type b combination vaccine (HIB); measles, mumps, and rubella vaccine (MMR); pneumococcal conjugate vaccine (PCV); diphtheria, tetanus, pertussis, inactivated poliovirus, haemophilus influenzae type b, and hepatitis B combination vaccine (DTaP-HBV-IPV/Hib vaccine); diphtheria, tetanus, pertussis, inactivated poliovirus, and haemophilus influenzae type b combination vaccine (DTaP-IPV/Hib vaccine).

**Table 2 vaccines-12-00429-t002:** Table on the forecast of vaccine refusals for preventable infections included in the NIS RK.

Vaccine Type	Years		Model Parameters		
	2026 Rate (95% CI)	2030 Rate (95% CI)	Type of Model	Alpha (Level)	
				t	*p*-Value
BCG	160.9 (−71.2; 393.1)	160.9 (−167.3; 489.3)	Simple	30.0	0.01
OPV	88.2 (−34.6; 211.1)	88.2 (−85.4; 262.0)	Simple	28.07	0.02
Hepatitis B	181.5 (−85.7; 448.7)	181.5 (−196.2; 559.3)	Simple	29.72	0.01
MMR	68.7 (−70.4; 207.8)	68.7 (−128.0; 265.4)	Simple	29.89	0.01
PCV	97.6 (−36.5; 231.9)	97.6 (−70.6; 266.0)	Simple	25.33	0.04
DTaP-HBV-IPV/Hib vaccine	88.8 (−63.3; 241.0)	88.8 (−101.9; 279.6)	Simple	26.04	0.04
DTaP-IPV/Hib vaccine	79.7 (−52.7; 212.2)	79.7 (−86.3; 245.8)	Simple	26.16	0.04

95% Confidence Interval (95% CI); bacille Calmette–Guérin (BCG); oral polio vaccine (OPV); diphtheria, tetanus, and pertussis (DTaP); hepatitis B vaccine (HBV); haemophilus influenzae type b combination vaccine (HIB); measles, mumps, and rubella vaccine (MMR); pneumococcal conjugate vaccine (PCV); diphtheria, tetanus, pertussis, inactivated poliovirus, haemophilus influenzae type b, and hepatitis B combination vaccine (DTaP-HBV-IPV/Hib vaccine); diphtheria, tetanus, pertussis, inactivated poliovirus, and haemophilus influenzae type b combination vaccine (DTaP-IPV/Hib vaccine).

**Table 3 vaccines-12-00429-t003:** An interrupted time series analysis was conducted to examine variations in the number of vaccine refusals before and after the introduction of the phased mass vaccination of the population against COVID-19 in Kazakhstan, starting from the beginning of February 2021.

Model Component	Type of Model	Stationary R^2^	Estimate (PPC)	*p* Value
BCG	Simple seasonal	0.610	16.807	0.0001
OPV	Simple seasonal	0.665	4.765	0.001
Hepatitis B	Simple seasonal	0.565	22.235	0.0001
MMR	Simple seasonal	0.586	17.435	0.0001
PCV	Simple seasonal	0.709	24.537	0.0001
DTaP-HBV-IPV/Hib vaccine	Winters additive	0.770	16.130	0.14
DTaP-IPV/Hib vaccine	Simple seasonal	0.767	17.311	0.001

Percentage point change (PPC); bacille Calmette–Guérin (BCG); oral polio vaccine (OPV); diphtheria, tetanus, and pertussis (DTaP); hepatitis B vaccine (HBV); haemophilus influenzae type b combination vaccine (HIB); measles, mumps, and rubella vaccine (MMR); pneumococcal conjugate vaccine (PCV); Diphtheria, tetanus, pertussis, inactivated poliovirus, haemophilus influenzae type b, and hepatitis B combination vaccine (DTaP-HBV-IPV/Hib vaccine); diphtheria, tetanus, pertussis, inactivated poliovirus, and haemophilus influenzae type b combination vaccine (DTaP-IPV/Hib vaccine).

## Data Availability

The datasets utilized or examined in the present study can be obtained from the corresponding author upon reasonable request.

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
