# Peer review of "The Association between Parental Child Vaccination Refusal Rate and the Impact of Mass Vaccination against COVID-19 in Kazakhstan: An Interrupted Time Series Analysis with Predictive Modelling of Nationwide Data Sources from 2013 to 2022"

_vaccines, 2024, doi:10.3390/vaccines12040429_

Round 1
Reviewer 1 Report
Comments and Suggestions for Authors
The objectives of this study are fairly explored.
The lengthy introduction provides clear information about the importance of vaccination and Kazakhstan's situation regarding vaccines.
In the "Results" section, it was interesting to note that there was a significant decrease in parents' distrust of combined forms of vaccines. I suggest adding more details on this aspect.
Regarding "Figure 1," it would be good to include a legend with colors to better and more easily visualize the results. Also, it might be useful to add the names of the regions on the map instead of numbers. The resulting map would be easier and more immediately understandable.
The article lacks some aspects relating to people who are partially or totally incapable of giving their consent; I also suggest adding further references on the topic (e.g: doi: 10.1163/15718093-bja10108; doi: 10.3390/vaccines9050429; doi.org/10.3390/healthcare11121793) in the Discussion section.
Comments on the Quality of English LanguageA minor revision of some English expressions is necessary.
Author Response
|
Referee 1 |
Changes by the authors |
|
The objectives of this study are fairly explored. |
Thank you for taking time to review our manuscript and for your thoughtful comments.
We considered all changes proposed by you and highlighted them in yellow. |
|
The lengthy introduction provides clear information about the importance of vaccination and Kazakhstan's situation regarding vaccines. |
Thank you! |
|
In the "Results" section, it was interesting to note that there was a significant decrease in parents' distrust of combined forms of vaccines. I suggest adding more details on this aspect.
|
Thank you for your comment. During our investigation, we identified a notable trend pertaining to the reduction in parental distrust towards combination vaccines compared to monovalent ones. We endeavored to describe this finding in the Results section as follows:
In most cases, there is a general trend of growth in the first half of the period (2013-2016), followed by stabilization or a reduction in refusal cases. Concerning certain vaccines (e.g., BCG and Hepatitis B), a significant increase in vaccine refusals is observed, while for others (DTaP-HBV-IPV/Hib and DTaP-IPV/Hib vaccines), there is a decline. However, despite the slight increase in refusals for vaccines such as DTaP and Hib, administered separately until 2016, there is a significant decrease in parental mistrust toward the combined forms of DTaP-HBV-IPV/Hib and DTaP-IPV/Hib vaccines, with a relative reduction of 3.38% (95% CI -16.08 -11.26%) and 3.99% (95% CI -16.25 - 10.07%), respectively. |
|
Regarding "Figure 1," it would be good to include a legend with colors to better and more easily visualize the results. Also, it might be useful to add the names of the regions on the map instead of numbers. The resulting map would be easier and more immediately understandable. |
Done. We have made adjustments to the formatting of Figure 1 in the manuscript:
|
|
The article lacks some aspects relating to people who are partially or totally incapable of giving their consent. I also suggest adding further references on the topic (e.g: doi: 10.1163/15718093-bja10108; doi: 10.3390/vaccines9050429; doi.org/10.3390/healthcare11121793) in the Discussion section
|
According to the vaccination procedure in Kazakhstan, vaccinations are carried out after obtaining informed consent from parents or legal representatives, either in paper or electronic form (https://adilet.zan.kz/eng/docs/P2000000612). Without obtaining informed consent, the procedure is not carried out. Therefore, this issue was not covered in the article.
The following clarification was added to the Introduction section: According to the vaccination procedure in Kazakhstan, vaccinations are administered subsequent to obtaining informed consent from parents or legal representatives, either in paper or electronic form. The procedure is not carried out without obtaining informed consent
|
Reviewer 2 Report
Comments and Suggestions for Authors
1. Do the authors have data on various disease reports during the same period? It may be more meaningful to find a relationship between vaccination and disease outbreaks.
2. Did the parents refuse vaccination without full vaccination? Or refuse a vaccine? It is important to distinguish this for this study. But the author's description in the article is vague.
3. In figure 1. What are the reasons for the significantly up trend of vaccination refusal in several western provinces? Is it something appeared irresistible or by human? What's the point for other provinces?
4. Less data, less importance of the article.
Author Response
|
Referee 2 |
Changes by the authors |
|
Do the authors have data on various disease reports during the same period? It may be more meaningful to find a relationship between vaccination and disease outbreaks
|
Thank you for considering our manuscript for publication in the Vaccines and for your thoughtful comments.
We have official data on morbidity for the specified periods. However, during the analysis of studies in the country, several studies [1–4] and sources [5,6], were found that provide official data in open access and analyze the coverage of vaccination and outbreaks of diseases. The aim of our study was to assess past trends in refusals of mandatory childhood vaccination in Kazakhstan for the period 2013-2022, forecast until 2030, and evaluate the impact of mass vaccination against COVID-19 on the refusal rates of childhood vaccination, as there is a lack of sufficient predictive studies on the hesitancy of childhood vaccination refusals in the country. |
|
2. Did the parents refuse vaccination without full vaccination? Or refuse a vaccine? It is important to distinguish this for this study. But the author's description in the article is vague
|
Thank you for your comment. We have addressed all the proposed amendments and highlighted them in yellow.
The data were compiled based on official statistics provided by the Committee of Sanitary and Epidemiological Control of the Ministry of Health and the National Statistics Bureau of the Agency for Strategic Planning and Reforms of the Republic of Kazakhstan and were divided by each type of vaccine. In other words, in the article, the data were summarized depending on refusals for all types of vaccines specified in the National Immunization Schedule. If parents refused all types of vaccines, then this data reflected refusals for all types of vaccines. If refusals concerned only certain types of vaccines, data were presented only for those specific vaccines. To provide a more detailed understanding of this aspect, we have added the following passage in the Methodology section:
In other words, the data in the article revealed the overall summarized indicator of refusals from childhood vaccinations, depending on refusals for each type of vaccine established by the National Immunization Schedule separately. |
|
In Figure 1. What are the reasons for the significantly up trend of vaccination refusal in several western provinces? Is it something appeared irresistible or by human? What's the point for other provinces?
|
Due to the retrospective design of our study, the research objective did not encompass the analysis of the causal relationship between vaccine refusal and vaccination coverage in Kazakhstan across regions. However, it is important to note some regional peculiarities in the western part of the country. During the National Comprehensive Sociological Study aimed at examining the knowledge, attitudes, and practices among parents of children aged 0-18 months, as well as healthcare workers involved in the child vaccination process, several factors influencing the refusal rates in the specified region were identified: low level of awareness, presence of distrust in the healthcare system due to the lack of communication skills among medical personnel, and presence of stereotypes and concerns regarding the composition and quality of vaccines. According to the research findings, the percentage of informed parents in the Mangystau region is 55.4%, which is on average 10-15% lower than that of residents in other regions. Additionally, substantial mistrust towards Kazakhstani healthcare was predominantly observed among surveyed residents of the Turkestan (27.4%) and Aktobe (25.7%) regions. Issues with data collection at the final stage of the study in rural areas of the Turkestan region, arising for unclear reasons, indicate regional peculiarities and the necessity to consider these factors in the development of preventive measures. Furthermore, the dissemination of myths and stereotypes about vaccine quality, composition, and impact on children's bodies played a significant role in spreading false scientifically unfounded information, distorting the real situation. The development of communication channels, availability, and openness of unofficial information transmission channels, often with emotional bias, also played a role in this process. Thus, the mentioned factors could have influenced the increase in vaccine refusal rates in these regions.
To discuss this issue further, we have added the following passage in Discussion section:
In accordance with the findings of previous studies, the percentage of informed parents in the Mangystau region was on average 10-15% lower than that of residents in other regions. Additionally, it was observed that fundamental mistrust towards Kazakhstani healthcare is most prevalent among respondents from the Turkestan (27.4%) and Aktobe (25.7%) regions, highlighting the imperative for increased attention from the healthcare sector in these areas [25].
|
|
4. Less data, less importance of the article. |
Thank you for the evaluation of our manuscript and recommendations made to improve the quality of it. We sincerely hope that our responses to all comments have been thorough and that we have fully explored the research topic. |
References
- Measles in Kazakhstan | UNICEF Available online: https://www.unicef.org/kazakhstan/en/reports/measles-kazakhstan (accessed on 3 April 2024).
- Sociological Research Report on Childhood Immunisation | UNICEF Available online: https://www.unicef.org/kazakhstan/en/reports/sociological-research-report-childhood-immunisation (accessed on 3 April 2024).
- Behaviour Insights Research on Drivers Influencing Immunization-Related Behaviours in Kazakhstan | UNICEF Kazakhstan Available online: https://www.unicef.org/kazakhstan/en/reports/behaviour-insights-research-drivers-influencing-immunization-related-behaviours-kazakhstan (accessed on 3 April 2024).
- Abeev, A.; Zhylkibayev, A.; Kamalova, D.; Kusheva, N.; Nusupbaeva, G.; Tleumbetova, N.; Smagul, M.; Beissenova, S.; Aubakirova, S.; Kassenova, Z.; et al. Epidemiological Outbreaks of Measles Virus in Kazakhstan during 2015. Jpn. J. Infect. Dis. 2018, 71, 354–359, doi:10.7883/YOKEN.JJID.2017.565.
- Percentage of Children Covered by Preventive (Prophylactic) Vaccinations Available online: https://bala.stat.gov.kz/en/ohvat-detej-profilakticheskimi-privivkami (accessed on 30 October 2023).
- WHO Immunization Data Portal - European Region Available online: https://immunizationdata.who.int/dashboard/regions/european-region/KAZ (accessed on 3 April 2024).
Reviewer 3 Report
Comments and Suggestions for Authors
This is a nicely written study however it would be helpful to explore reasons for vaccine hesitancy in more detail. Do you have any proxy measures for population deprivation levels in each of the regions? The raw mortality rate or the age standardized mortality rate (ASMR) are two possible indicators. Older populations will have a higher raw mortality rate and could be less inclined to vaccinate children. Populations with a high ASMR will experience poor health/poverty and may likewise neglect childhood vaccination. Additional measures of social deprivation or regional education levels may be available. If you could explore any regional associations this would serve to make this paper of wider relevance.
To my mind the reduced uptake of BCG is especially troubling. This is because the BCG has widely recognized nonspecific effects against all-cause mortality. These wider beneficial effects have also been observed for other common vaccines. I would therefore suggest a section in the introduction highlighting this.
If I may also suggest that a degree of vaccine hesitancy regarding COVID vaccination may be wise. My own unpublished work shows that the COVID vaccine performed best at reducing all-cause mortality specifically at age 80-84 and the performance declined either side. I have some evidence that COVID vaccination in children led to an increase in all-cause mortality, especially at age 10-14, i.e., around puberty.
Other research suggests that the nonspecific effects of influenza vaccination (in adults) has variable effects against all-cause excess winter mortality and in some years countries with higher influenza vaccination rates have higher excess winter mortality than those with the lowest rates of vaccination.
On a positive side these appear to the only two vaccines where nonspecific effects against all-cause mortality seem to affect some groups. This is perhaps due to the rapidly mutating surface coat of these two pathogens where an antigenic mismatch can occur between locally circulating variants/clades and the antigen mix in the vaccine.
Clearly the less educated populations may struggle to understand how these two vaccines can deliver specific benefit against its target pathogen yet at the same time cause adverse nonspecific effects. At the same time all named childhood vaccines are highly recommended and relevant with the added benefit of beneficial nonspecific effects.
Lower education levels are a well recognized factor in vaccine hesitancy and hence the suggestion that this may correlate at regional level.
The curious jump up/down in refusal rates for certain vaccines in the earlier years requires some comment and should be investigated at regional level.
Refusal numbers should probably be expressed as rates of refusal per 100,000 children, in each region.
In conclusion, some extra analysis and wider literature search would be of great benefit to this paper.
Author Response
|
Referee 3 |
Changes by the authors |
|
This is a nicely written study however it would be helpful to explore reasons for vaccine hesitancy in more detail
|
Thank you for the evaluation of our manuscript and recommendations made to improve the quality of it.
We have addressed all the proposed amendments and highlighted them in yellow.
Due to the utilization of official data on vaccine hesitancy and the retrospective design of the study, the aim did not encompass an analysis of the primary reasons for vaccination hesitancy in Kazakhstan. This stands as one of the limitations of our study. However, we acknowledge the significance and necessity of conducting analysis to delineate the causal relationship between factors of vaccine refusal and vaccination coverage, considering regional characteristics within the country in future studies. Furthermore, we wish to highlight the findings of a recent National Comprehensive Sociological Study aimed at examining the knowledge, attitudes, and practices among parents, along with healthcare workers involved in the vaccination process for children in Kazakhstan. These findings were discussed in the Discussion section. Despite the myriad reasons for vaccination refusal, certain regional characteristics have been identified in specific parts of the country, manifesting in increased rates of refusal - including low levels of awareness, distrust of the healthcare system due to inadequate communication from medical personnel, as well as the prevalence of stereotypes and concerns regarding the composition and quality of vaccines.
|
|
Do you have any proxy measures for population deprivation levels in each of the regions? The raw mortality rate or the age standardized mortality rate (ASMR) are two possible indicators. Older populations will have a higher raw mortality rate and could be less inclined to vaccinate children. Populations with a high ASMR will experience poor health/poverty and may likewise neglect childhood vaccination. Additional measures of social deprivation or regional education levels may be available. If you could explore any regional associations this would serve to make this paper of wider relevance.
|
The data presented in the article has been scrutinized using official statistics provided by the Committee of Sanitary and Epidemiological Control of the Ministry of Health and the National Statistics Bureau of the Agency for Strategic Planning and Reforms of the Republic of Kazakhstan. Information is compiled and collected according to an approved reporting format by healthcare professionals, including nurses and statisticians, contingent upon the level of healthcare provision (village, district, city, region, center), facilitated through collaboration among healthcare organizations. Data is gathered from a targeted sample, in our case, parents/guardians. Unfortunately, we did not find proxy measures for population deprivation levels in Kazakhstan. However, we hope to identify such indicators in our country in the future. |
|
To my mind the reduced uptake of BCG is especially troubling. This is because the BCG has widely recognized nonspecific effects against all-cause mortality. These wider beneficial effects have also been observed for other common vaccines. I would therefore suggest a section in the introduction highlighting this.
|
Done. To discuss this issue further, we have added the following passage in the Introduction as follows:
It is also important to note the nonspecific impact of mandatory childhood vaccination on reducing all-cause mortality. Several studies have suggested broader beneficial effects of these vaccines through the induction of innate immune memory, which has a beneficial effect on survival, especially in children with weakened immune systems. Therefore, decreased vaccination coverage and increased hesitancy could lead to serious public health problems in the long term [2–6]. |
|
If I may also suggest that a degree of vaccine hesitancy regarding COVID vaccination may be wise. My own unpublished work shows that the COVID vaccine performed best at reducing all-cause mortality specifically at age 80-84 and the performance declined either side. I have some evidence that COVID vaccination in children led to an increase in all-cause mortality, especially at age 10-14, i.e., around puberty. Other research suggests that the nonspecific effects of influenza vaccination (in adults) has variable effects against all-cause excess winter mortality and in some years countries with higher influenza vaccination rates have higher excess winter mortality than those with the lowest rates of vaccination. On a positive side these appear to the only two vaccines where nonspecific effects against all-cause mortality seem to affect some groups. This is perhaps due to the rapidly mutating surface coat of these two pathogens where an antigenic mismatch can occur between locally circulating variants/clades and the antigen mix in the vaccine. Clearly the less educated populations may struggle to understand how these two vaccines can deliver specific benefit against its target pathogen yet at the same time cause adverse nonspecific effects. At the same time all named childhood vaccines are highly recommended and relevant with the added benefit of beneficial nonspecific effects. Lower education levels are a well recognized factor in vaccine hesitancy and hence the suggestion that this may correlate at regional level. The curious jump up/down in refusal rates for certain vaccines in the earlier years requires some comment and should be investigated at regional level.
|
Thank you for your comment! We fully agree with your opinion. It is important to note some regional peculiarities in the western and southern regions of the country. During the National Comprehensive Sociological Study aimed at examining the knowledge, attitudes, and practices among parents of children aged 0-18 months, as well as healthcare workers involved in the child vaccination process, several factors influencing the refusal rates in the specified region were identified: low level of awareness, presence of distrust in the healthcare system due to the lack of communication skills among medical personnel, and presence of stereotypes and concerns regarding the composition and quality of vaccines. According to the research findings, the percentage of informed parents in the Mangystau region is 55.4%, which is on average 10-15% lower than that of residents in other regions. Additionally, substantial mistrust towards Kazakhstani healthcare was predominantly observed among surveyed residents of the Turkestan (27.4%) and Aktobe (25.7%) regions. Issues with data collection at the final stage of the study in rural areas of the Turkestan region, arising for unclear reasons, indicate regional peculiarities and the necessity to consider these factors in the development of preventive measures. Furthermore, the dissemination of myths and stereotypes about vaccine quality, composition, and impact on children's bodies played a significant role in spreading false scientifically unfounded information, distorting the real situation. The development of communication channels, availability, and openness of unofficial information transmission channels, often with emotional bias, also played a role in this process. The introduction of COVID-19 vaccination has posed a number of challenges for the healthcare system in Kazakhstan, particularly in light of existing issues with declining coverage of other mandatory vaccinations. The negative experience of introducing the HPV vaccine, fear of potential vaccination consequences, and beliefs about the unnecessary nature of influenza vaccination make this process even more complex All of this underscores the need to improve current measures. Thus, the mentioned factors could have influenced the increase in vaccine refusal rates in these regions could cause jump up/down in refusal rates for vaccines.
To discuss this issue further, we have added the following passage in the Discussion section:
In accordance with the findings of previous studies, the percentage of informed parents in the Mangystau region was on average 10-15% lower than that of residents in other regions. Additionally, it was observed that fundamental mistrust towards Kazakhstani healthcare is most prevalent among respondents from the Turkestan (27.4%) and Aktobe (25.7%) regions, highlighting the critical need for increased at-tention from the healthcare sector in these areas [25]
|
|
Refusal numbers should probably be expressed as rates of refusal per 100,000 children, in each region.
|
Thank you for pointing out the error. We corrected it rates of refusal per 100,000 children, in each region and highlighted them in yellow as follows:
To visualize regional changes in the dynamics of parental/guardian vaccination refusals per 100,000 children aged 0-14, maps of Kazakhstan were created for three temporal periods: 2013-2016, 2017-2019, and 2020-2022 using ArcMap version 10.8.1 by ESRI.
It is worth noting a sharp increase in the refusal level in the Aktobe region by 13.9 times since 2013 (from 123.5 to 1726.1) and in the Atyrau region by 4.29 times (from 244.1 to 1048.6). The lowest refusal rates in all three periods are observed in the East Kazakhstan, Zhambyl, and Kyzylorda regions. However, in the Kyzylorda region, despite comparatively low refusal rates compared to other regions, a significant increase in rates has been observed over the last 10 years (from 45.3 to 200.3). Stable rates are maintained in the North Kazakhstan region from 2017 to 2022. A decrease in rates is also observed in three regions from 2017 to 2022: Almaty (from 246.6 to 220.4), West Kazakhstan (from 579.2 to 481.1), and Zhambyl region (from 155.4 to 141.2).
Notably, there is a sharp increase in refusal rates in the Aktobe region by 13.9 times since 2013 (from 123.5 to 1726.1) and in the Atyrau region by 4.29 times (from 244.1 to 1048.6). |
|
In conclusion, some extra analysis and wider literature search would be of great benefit to this paper. |
Thank you for your feedback and recommendations. We sincerely hope that our responses to all comments have been thorough and that we have fully explored the research topic. |
Reviewer 4 Report
Comments and Suggestions for Authors
First of all, I would like to congratulate you for this article. It is very much valid and the need of the hour to address vaccine refusals particularly in the pediatric age group as it is controlled by parent's own innate fears or because of their own misinformation.
A few comments though of my own:
1.) Was the educational background of the parents taken into consideration, particularly in the regions of Aktobe and Atyrau where the refusal rates of vaccines were particularly high?
2.) Any comments on general mistrust with governmental agencies with regards to vaccines especially due to past failures/side effects
3.) Is information about the vaccines particularly with regards to safety percolated to the grass root level of the public?
4.) Vaccinations dropped during the COVID pandemic due to obvious reasons, but vaccine uptake in your study particularly dropped used against respiratory infections viz. measles, pertussis, diphtheria etc. Did the rates of these infections drop down in general due to masking, social isolation, hand washing etc. during the COVID period which gave a false sense of security among parents, hence affecting the uptake of vaccines.
Author Response
|
Referee 4 |
Changes by the authors |
|
First of all, I would like to congratulate you for this article. It is very much valid and the need of the hour to address vaccine refusals particularly in the pediatric age group as it is controlled by parent's own innate fears or because of their own misinformation. A few comments though of my own |
Thank you for considering our manuscript for publication in the Vaccines and for your thoughtful comments.
We considered all changes proposed by you and highlighted them in yellow. |
|
Was the educational background of the parents taken into consideration, particularly in the regions of Aktobe and Atyrau where the refusal rates of vaccines were particularly high?
|
The information presented in the article has been analyzed using official statistics provided by the Committee of Sanitary and Epidemiological Control of the Ministry of Health and the National Statistics Bureau of the Agency for Strategic Planning and Reforms of the Republic of Kazakhstan. Data is filled out and collected based on an approved reporting form by healthcare professionals, including nurses and statisticians, depending on the level of healthcare provided (village, district, city, region, center), through the interaction of healthcare organizations. Information is collected from a targeted sample (in our case, parents/guardians), regardless of educational level. |
|
Any comments on general mistrust with governmental agencies with regards to vaccines especially due to past failures/side effects
|
Several national studies have been conducted in Kazakhstan to identify behavioral factors influencing attitudes towards immunization. Mistrust of health authorities is one of the significant factors influencing parents' decisions regarding childhood vaccination.
To discuss this issue further, we have added the following passage:
Contributing factors to this trend include the global COVID-19 pandemic, limited knowledge about vaccines and vaccination sources and presence of distrust in the healthcare system due to the lack of communication skills among medical personnel |
|
Is information about the vaccines particularly with regards to safety percolated to the grass root level of the public?
|
Information about vaccination is available to all levels of the public, but recent data from the National Survey in Kazakhstan on knowledge, attitudes, and practices among parents and health workers showed that the population has a superficial understanding of the process of childhood vaccination, its benefits, and risks. Although parents are generally familiar with the vaccination procedure, many details remain unknown to them, such as the names of the vaccines, their composition, countries of origin, sequence of vaccinations, and other aspects. The exception is those who refuse vaccination on principle; such people are usually better informed, especially regarding possible risks, which can often be distorted or exaggerated. |
|
Vaccinations dropped during the COVID pandemic due to obvious reasons, but vaccine uptake in your study particularly dropped used against respiratory infections viz. measles, pertussis, diphtheria etc. Did the rates of these infections drop down in general due to masking, social isolation, hand washing etc. during the COVID period which gave a false sense of security among parents, hence affecting the uptake of vaccines.
|
Due to the retrospective design of our study, we did not explore the cause-and-effect relationship between disease incidence and preventive measures against COVID-19. Recent reviews on COVID-19 (Causey et al., 2021) have highlighted a decline in vaccination coverage for respiratory infections worldwide, particularly with regards to third doses of the diphtheria, tetanus, and pertussis vaccine (DTaP3) and the first dose of the measles-containing vaccine (MCV1). However, we acknowledge the importance of conducting in-depth qualitative studies that would help identify the relationship between these two indicators. |
Round 2
Reviewer 2 Report
Comments and Suggestions for Authors
The author's reply is quite pertinent and has been revised according to expert opinions.
Author Response
Dear reviewer,
Thank you once again for taking the time to evaluate our manuscript and for providing constructive feedback to enhance its quality.
Reviewer 3 Report
Comments and Suggestions for Authors
Thank you for addressing the comments. As one final point can you make some comments about the large jump in refusal rates in Figure 3 between 2016 and 2018. There must be a reason and it could potentially distort the interpretation of the trends. This bulge looks odd. Were there any major differences between Regions?
Author Response
|
Referee 3 |
Changes by the authors |
|
Thank you for addressing the comments. As one final point can you make some comments about the large jump in refusal rates in Figure 3 between 2016 and 2018. There must be a reason and it could potentially distort the interpretation of the trends. This bulge looks odd. Were there any major differences between Regions?
|
Thank you for the evaluation of our manuscript and recommendations made to improve the quality of it.
We considered all changes proposed by you and highlighted them in yellow.
We concur with your observation that there is a remarkable trend of increasing vaccine refusal rates starting from 2016. It is noteworthy that this period has been one of the most challenging and demanding for healthcare in Kazakhstan due to the rise in infectious diseases [1,2]. The conducted research indicates a significant surge in vaccine refusals during this period, peaking in 2016 with 43,949 cases. Despite a slight decrease in 2017, the level of refusals remained high, exhibiting fluctuations in subsequent years. The primary reasons for vaccine hesitancy include personal religious beliefs, media influence, and distrust in vaccines. Additionally, it is important to highlight the regional peculiarities in the western and southern regions of the country, where vaccine refusal rates are increasing due to religious beliefs, necessitating intensified attention and preventive measures.
To discuss this issue further, we have added the following passage in the Discussion section:
The conducted study revealed a substantial increase in vaccine refusals during the observed period. The peak of this rise occurred in 2016, with the number of refusal cases reaching 43,949. Despite a slight decrease and variability in the subsequent years from 2017 to 2019, record-high values were recorded during and after the pandemic period: 42,282 cases in 2021 and 44,180 cases in 2022. There has also been a sharp rise in refusal rates in specific regions of Kazakhstan, notably in the Aktobe region (by 13.9 times) and Atyrau region (by 4.29 times). In accordance with the findings of previous research, regional disparities are observed in the western and southern regions of the country, where vaccine refusal continues to increase for several reasons, including low levels of awareness, religious beliefs, and distrust of vaccines. In the Mangystau region, the percentage of informed parents is on average 10-15% lower than in other regions. Additionally, it has been noted that fundamental distrust of Kazakhstani healthcare is most prevalent among respondents from the Turkestan (27.4%) and Aktobe (25.7%) regions, underscoring the need for increased attention from healthcare authorities in these areas.
|